# Development of an imputation model to recalibrate birth weights measured in the early neonatal period to time at delivery and assessment of its impact on size-for-gestational age and low birthweight prevalence estimates: a secondary analysis of a pregnancy cohort in rural Nepal

Elizabeth A Hazel [1], Luke C Mullany [1], Scott L Zeger [1], Diwakar Mohan,[1] Seema Subedi [1], James M Tielsch [2], Subarna K Khatry,[3] Joanne Katz [1]

For numbered affiliations see end of article.

**Correspondence to**
Dr Elizabeth A Hazel; ehazel1@jhu.edu

## ABSTRACT

**Objectives** In low-income countries, birth weights for home deliveries are often measured at the nadir when babies may lose up of 10% of their birth weight, biasing estimates of small-for-gestational age (SGA) and low birth weight (LBW). We aimed to develop an imputation model that predicts the 'true' birth weight at time of delivery.

**Design** We developed and applied a model that recalibrates weights measured in the early neonatal period to time=0 at delivery and uses those recalibrated birth weights to impute missing birth weights.

**Setting** This is a secondary analysis of pregnancy cohort data from two studies in Sarlahi district, Nepal.

**Participants** The participants are 457 babies with daily weights measured in the first 10 days of life from a subsample of a larger clinical trial on chlorhexidine (CHX) neonatal skin cleansing and 31 116 babies followed through the neonatal period to test the impact of neonatal massage oil type (Nepal Oil Massage Study (NOMS)).

**Outcome measures** We developed an empirical Bayes model of early neonatal weight change using CHX trial longitudinal data and applied it to the NOMS dataset to recalibrate and then impute birth weight at delivery. The outcomes are size-for-gestational age and LBW.

**Results** When using the imputed birth weights, the proportion of SGA is reduced from 49% (95% CI: 48% to 49%) to 44% (95% CI: 43% to 44%). Low birth weight is reduced from 30% (95% CI: 30% to 31%) to 27% (95% CI: 26% to 27%). The proportion of babies born large-for-gestational age increased from 4% (95% CI: 4% to 4%) to 5% (95% CI: 5% to 5%).

**Conclusions** Using weights measured around the nadir overestimates the prevalence of SGA and LBW. Studies in low-income settings with high levels of home births should consider a similar recalibration and imputation model to

## STRENGTHS AND LIMITATIONS OF THIS STUDY

⇒ A strength of this study is that we developed our model of early neonatal growth using longitudinal data of daily weight change.

⇒ Another strength is that the data used to develop this model was sampled from a similar population of infants in rural Nepal as was the study used to apply the model; although there may be differences due to secular trends, the median birth weights and neonatal mortality rates of the two studies are similar.

⇒ A further strength is that both studies included high-quality gestational age and weight measurements taken on a population-representation sample with relatively low levels of missing data.

⇒ A limitation of the study is that longitudinal data used to build the recalibration model did not have multiples or early neonatal deaths—babies that likely have different early growth patterns—although we are able to impute birth weight for multiples and neonatal deaths in the application model, it was based on measured weights only.

⇒ Another limitation is that the model may not be generalisable to low-income settings outside of South Asia where prevalence of intrauterine growth restriction is lower.

generate more accurate population estimates of small and vulnerable newborns.

## INTRODUCTION

In 2010, 32.4 million babies in low-income and middle-income countries (LMICs) were estimated to be small-for-gestational age (SGA), over a quarter of all live births and

these babies had an 83% higher risk of dying the first month of life.[1 2] The South Asia region has the highest prevalence of SGA in the world; a third of babies are born small (34%) compared with the global rate (19%).[3] SGA is defined as a birth weight-for-gestational age less than the 10th percentile of a standard infant reference population and severe SGA is less than the 3rd percentile. A low weight at birth (defined as babies born <2500 g regardless of gestational age) may result from a preterm birth (births <37 completed weeks gestational age), intra-uterine growth restriction (IUGR) or a combination of the two. The aetiology of IUGR is likely distinct from those born preterm, and perhaps even different from babies born early with IUGR. To better understand the epidemiology of low birthweight babies, practitioners and researchers have moved to monitoring birth weight by gestational age, although low birthweight prevalence is still appropriate to use in settings where gestational age measured by ultrasound in the first trimester (the gold standard) is rare.[4] Some babies will be on the smaller end of the curve naturally, but in LMICs where SGA estimates are high, it can be used as a proxy for IUGR prevalence. Large-for-gestational age (LGA) or babies with a birth weight >90th centile have associated health risks.[5 6] SGA, LGA and low birthweight prevalence in LMICs are essential indicators of population health.

Along with gestational age, an accurate birth weight measured close to the time of birth is critical for estimating SGA. This is because neonatal weight decreases for the first 2–3 days after birth (known as the *nadir* or period of minimum neonatal weight), before starting to increase. A systematic review of breastfed, healthy, singleton babies from high-income countries found the nadir of mean weight loss of 7%–8% from birth weight at day 3. Many neonates in these studies had weight loss >7% and up to 10% from birth weight for exclusively breastfed babies.[7]

Birth weight is typically measured shortly after delivery for babies born at health facilities. For babies born outside of facilities in LMICs, it may take 24–72 hours before a health worker can measure weight, often at the nadir of the infant's weight loss. Weights of babies taken during the nadir may overestimate SGA and low birthweight prevalence and underestimate LGA.

In this study, we aimed to generate estimates of SGA, LGA and low birth weight using an imputation model that extrapolates infant weights measured postdelivery (around the nadir) to a recalibrated weight at the time of birth and then uses those recalibrated birth weights to impute for babies missing a weight measurement.

## METHODS
### Data sources

This was a secondary data analysis of two pregnancy cohort datasets from rural Nepal (table 1). The first was derived from a cluster-randomised community-based trial examining the effect of chlorhexidine (CHX) full body wipe at delivery and CHX umbilical stump application on neonatal

**Table 1** Description of datasets used for analysis

| | Training data | Study data |
|---|---|---|
| Name | Chlorhexidine trial | Nepal Oil massage study (NOMS) |
| Setting | Sarlahi district, Nepal | Sarlahi district, Nepal |
| Years of data collection | 2002–05 | 2010–17 |
| Number of live births included in the analysis | 457 | 31 116 |
| Male gender (%) | 57.6 95% CI: 53.0 to 62.0 | 51.9 95% CI: 51.3 to 52.4 |
| Median gestational age at delivery (weeks) | 39.7 IQR: 37.8–41.2 | 39.4 IQR: 38.0–40.9 |
| Median birth weight (g) | 2706 IQR: 2480–2986 | 2740 IQR: 2450–3040 |
| Time of first weight measurement after birth (hours) | | |
| Median | 4.1 IQR: 3.1–5.2 | 15.2 IQR: 7.9–25.3 |
| Mean | 4.0 SD: 1.28 | 89.1 SD: 275.42 |
| Preterm, <37 weeks gestational age (%) | 14.9 95% CI: 11.9 to 18.5 | 15.7 95% CI: 15.3 to 16.1 |
| Health facility delivery (%) | 3.7 95% CI: 2.3 to 5.9 | 58.1 95% CI: 57.6 to 58.7 |
| Gestational age ascertainment | Last menstrual period | Last menstrual period |
| Instrument used to measure infant weight | Seca Digital Scale Model 727 and Tanita digital infant weight scale | Tanita digital infant weight scale |

NOMS, Nepal Oil Massage Study.

omphalitis and neonatal mortality.[8 9] From 2002 to 2005, pregnant women were recruited approximately in the sixth month of pregnancy and infants were followed up on 1, 3, 7, 10, 14, 21 and 28 days of age. The second was from another cluster-randomised community-based trial called the Nepal Oil Massage Study (NOMS) (clinicaltrials.gov #NCT01177111). From 2010 to 2017, pregnant women were enrolled through community surveillance and their newborns were randomised to receive postnatal massage using either sunflower or mustard seed oil to determine the impact of oil type on neonatal morbidity and mortality. Both studies took place in Sarlahi district in southern Nepal. In this area, less than half of the women have delivery in health facilities (40%), caesarean sections are very low (<1%) and generally 90% of infants are breast fed within the first day of delivery.[10]

In the CHX trial, a subset of infants (n=457) with first weight measured in the first 6 hours after delivery were purposively sampled for the longitudinal substudy, and then followed for daily weight measurements through the first 10 days of life. Gestational age was calculated using last menstrual period (LMP). Multiples and babies that did not survive the early neonatal period were not included in the CHX subsample of longitudinal data. We selected these data

to model early neonatal weight change since it drew from the same population (but earlier timepoint) as the NOMS study.

For the NOMS study, women in the study area were visited approximately every 5 weeks and asked about the timing of their LMP. Pregnant women were enrolled throughout the pregnancy (median gestational age at enrolment was 14.1 weeks) and some women contributed multiple pregnancies to the study data. Birth weight was collected at the first postnatal visit on all surviving liveborn infants. The distribution of visit timing is skewed right since 8% of the babies had their first postnatal visit after day 10 (table 1). Information on maternal age, parity, education, household wealth and infant sex and survival status were collected through interviews with the mother. In both studies, birth weight was measured by trained study staff using digital scales.

## Descriptive analysis

For the CHX longitudinal data, we calculated the median and IQR for nadir weight, time in days to reach the nadir and time in days to return to the birth weight in the 10-day period. To make these estimates, we developed a function that extends the predicted weights from time of delivery to a maximum observed time in units of days (within the 10-day period). The function then finds the time in days when the predicted weight achieves its minimum and when the weight returns closest to the birth weight. We plotted a survival curve for probability of return to birth weight within the 10 days. Babies that did not return to their birth weight in the 10-day period were censored.

For the NOMS data, we described the sample and analysed the pattern of missing weights and weights taken >72 hours after delivery to determine if babies with missing and non-missing weights differed by preterm status (<37 weeks), survival status through the neonatal period and mother's parity and educational level.

## Predicting birth weight

We imputed weight at birth in the NOMS dataset by estimating, then recalibrating from the conditional distribution of each child's birth weight (weight at t=0) given a single measurement at a known later time t>0 (case 1) or imputing given missing weight (case 2).

In both cases, the strategy had three steps: (1) use the longitudinal CHX study to estimate the dependence of birth weight on later weights and assume this relationship holds approximately for the NOMS data as well; (2) use the NOMS data to model the dependence of population average weight on age at measurement and other key predictors; (3) integrate (1) and (2) to obtain the optimal linear predictor of birth weights in NOMS given the single weight measurement on each child$_i$ at age $t_i$=j.

$$Y_{ij} = \beta_0 + \beta_1 X_{1ij} + \ldots \beta_p X_{pij} + u_{0i} + u_{1i}t_i + e_{ij}$$

For the first step, the CHX data were used to fit a linear mixed effect model with the repeated weights as the outcomes ($Y$ is the daily weight for $j$th of $n_i$ for each baby). The fixed effects $\left(X_{1ij} \ldots X_{pij}\right)$ included parity, maternal education/age, infant sex and gestational age at birth in weeks with a spline at 40 weeks. The random effects within baby include intercept $(u_{0j})$ and random slope $(u_{1i})$.

This model enabled us to estimate the conditional distribution of the unknown birth weight given one or more weights at later times, controlling for fixed effects. The mean of this distribution is the empirical Bayes estimate of birth weight. We applied the variance parameters from the training model to NOMS data to generate the distribution of the imputed birth weights for NOMS (box 1).

For the second step, using NOMS data, we regressed the observed weight ($w_t$) at age $t$ in hours on: a smooth function of maternal age and education in years, the neonate's survival status by day of age, infant sex, an indicator of preterm birth and whether the birth was a multiple or singleton, and whether it was the mother's first birth (parity) using a linear mixed effect model ($Y$ is the estimated weight for a $i$th woman's $j$th pregnancy that resulted in a live birth with a measured weight). Neonatal survival status was defined as a liveborn death in the first 28 days of life with the reference being children who survived the neonatal period (28 days after birth) or alive at the time they were lost to follow-up. We used this model to obtain the estimated weight $\left(\hat{w}_{it}^{NOMS}\right)$ for infant $i$ at their measurement time $t$ and their predicted birth weight $\left(\hat{W}_{i0}^{NOMS}\right)$ at $t$=0 (box 1).

In the final step (3), we applied the relationship between birth weight and subsequent weights derived in step (1) to the NOMS observed and predicted weights to obtain an improved prediction (recalibration) of the NOMS infant's birth weights (box 1). We generated m=5 datasets of the imputed birth weight.

We used these recalibrated weights to impute for infants with missing weight data (case 2). We fit a random forest

---

**Box 1  Formulas for calculating the empirical Bayes estimate of weight and its variance, t=0**

⇒ $w_{it}$=measured weight for child $i$ at time $t>0$, Nepal Oil Massage Study (NOMS).

⇒ $\hat{W}_{it}^{NOMS}$=predicted weight at actual measurement time ($t>0$), NOMS.

⇒ $\hat{w}_{i0}^{NOMS}$=predicted weight, $t$=0 from the linear mixed effect model, NOMS.

⇒ $G$=2×2 variance-covariance matrix of random effects in the linear mixed effects model, chlorhexidine (CHX).

⇒ $\sigma2$=residual variance from the linear mixed effects model, CHX.

⇒ $Z_i$=$n_i$×2 random effects design matrix where $n_i$ is the number of ages which are being used to predict the birth weight, here $n_i$=1 and $Z_i$=(1,age), NOMS.

⇒ $v_i$=$[Z_iG t(Z)]_{1,2}$=covariance of the linear combination of the random effects that contribute to the birth weight prediction, NOMS, CHX.

⇒ $\tilde{w}_{i0}^{NOMS}$=empirical Bayes estimate of birth weight =$\hat{W}_{i0}^{NOMS} + v_i/\left(v_i + \sigma^2\right)\left(w_{it} - \hat{W}_{it}^{NOMS}\right)$

⇒ $Var\left(\tilde{w}_{i0}^{NOMS}\right) = \left(G_{11} + \sigma^2\right) - v_i^2/\left(v_i + \sigma^2\right)$.

model to the recalibrated birth weight dataset with baby's gender, gestational age at delivery, multiple/singleton, neonatal survival status and maternal parity, education and age. Using this model, we then imputed birth weight for babies with no weights measured and randomly generated a m=5 dataset for imputation. Infants with missing parity, survival status, maternal age/education, gestational age, singleton/multiples or sex data were excluded from the imputation. All modelling for case 1 and case 2 were performed in R.[11]

### Comparing size-for-gestational age and low birth weight using raw and imputed weights

We generated appropriate-for-gestational age (AGA) (10th–90th centiles), SGA (<10th centile) and LGA (>90th centile) using Intergrowth-21 standards extrapolated up to 22+0 to 44+6 weeks GA (personal communication, Eric Ohuma, London School of Hygiene and Tropical Medicine, September 2021) (original model was 24+0 to 42+6 weeks).[12] We generated imputed estimates of AGA, SGA, LGA and low birthweight (<2500 g) and very low birthweight (<1500 g) prevalence with multiple imputation (m=5) in Stata V.16.1.[13] The main outcome was the prevalence estimate comparison using the imputed birth weight versus measured weight within 72 hours. As a sensitivity analysis, we compared the proportion of term and preterm babies, female and male gender babies and the neonatal mortality rate using the dataset with the raw birth weight versus the imputed dataset to see whether babies are being included or excluded with the imputation.

### Patient and public involvement

The patients and public were not involved in the design, conduct, reporting or dissemination of this research. Accurately measuring the health and nutritional status of infants is important to clinicians, researchers and families in Nepal, however, we did not seek the public's priorities, experience or preferences when designing this research question. We did not contact the original participants in the two studies used in this secondary data analysis.

### RESULTS

The median birth weight in the CHX trial (training data) was 2726 (IQR 2490–2976) (table 2). The median time of nadir was 2 days and the median weight at the nadir was 2609 (IQR 2370–2859); a relative weight loss of 4% or 117 g. Median return to birth weight was 5 days and 84 infants did not regain their birth weight within the 10 days (figure 1).

From the model outputs, we found babies born to mothers with no previous births were estimated to have a lower than average birth weight and baby girls born at a greater gestational age born to older mothers were estimated to have higher birth weights on average (online supplemental table A1). There was minimal impact of maternal education on birth weight adjusted by other

**Table 2** Neonatal growth during the first 10 days of life, chlorhexidine trial (n=4148 weight measurements on 456 infants)

|  | Median (IQR) |
| --- | --- |
| Birth weight (g) | 2726.3 (2490.2–2975.9) |
| Nadir weight (g) | 2609.0 (2369.5–2859.1) |
| Time to reach nadir (days) | 2.1 (2.0–2.2) |
| Time to return to birth weight (days) | 4.5 (3.8–5.7) |
| Number of babies that did not return to birth weight within the 10 days | 84 |

covariates. We compared this model with others excluding different maternal level factors and the full model has the best fit with small differences in the *Akaike Information Criterion* (online supplemental table A2).

There were 31 116 live births from 22 495 women in the NOMS dataset (table 3). We imputed a birth weight on 99% of the live births (n=30 996) with complete covariate and gestational age data. Of those live births, 3863 were missing a weight measurement and we were able to impute a birth weight for 98% (n=3769). The pattern of missing birth weights was not random (online supplemental table A3). Infants without a measured weight were more likely to be born to nulliparous mothers with more education. Infants who were preterm were more likely to have a missing birth weight (16% preterm compared with 12% full term) and over half of deceased infants did not have a weight measured (62%) compared with a 10th of infants who survived during the neonatal period (11%). Those infants with a weight measured after 72 hours showed a similar pattern (online supplemental table A3).

For the NOMS imputation, at day 0 the imputed birth weights (mean 2769 g, 95% CI: 2763 to 2775 g) were on average 50 g higher than the measured weights (mean 2719 g, 95% CI: 2713 to 2725 g) and on average 131 g higher on days 1–3 (figure 2). During days 4–9 and 10+, when the baby begins to regain and surpass birth weight, the recalibration reduces the weights back down (on

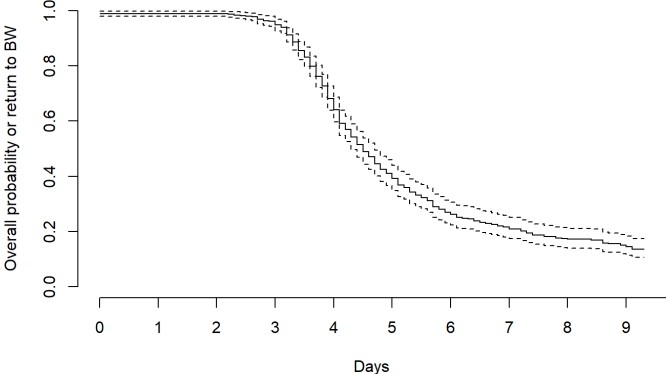

**Figure 1** Probability of return to birth weight (BW) and 95% CIs, chlorhexidine trial. Babies that did not return to their BW in the 10-day period were censored.

**Table 3** Description of covariate data for all live births, NOMS study

| | All live births (n=31 116) % (n) |
|---|---|
| Maternal age at LMP (years) | |
| <18 | 15.5 (4823) |
| 18–35 | 82.3 (25 622) |
| >35 | 2.2 (671) |
| Missing | 0 |
| Maternal education | |
| No school | 67.0 (20 855) |
| 1–5 years | 8.5 (2638) |
| >5 years | 24.4 (7595) |
| Missing | 0.1 (28) |
| Parity (live birth or stillbirth) | |
| No previous births | 31.5 (9794) |
| 1 | 27.4 (8531) |
| 2 | 19.6 (6108) |
| 3 | 10.8 (3370) |
| 4+ | 10.1 (3147) |
| Missing | 0.5 (166) |
| Gestational age (weeks) | |
| Term (≥37 to <45) | 84.3 (26 243) |
| Moderate-to-late preterm (≥32 to <37) | 13.7 (4262) |
| Early preterm (≥28 to <32) | 1.6 (501) |
| Extremely preterm (<28) | 0.4 (110) |
| Missing | 0 |
| Infant sex | |
| Female | 48.0 (14 925) |
| Male | 51.7 (16 099) |
| Missing | 0.3 (92) |
| Multiple births | |
| Singleton | 98.4 (30 616) |
| Twin/Triplets | 1.6 (500) |
| Missing | 0 |
| Infant survival status | |
| Alive at 28 days | 96.6 (30 041) |
| Lost to follow-up | 3.5 (1076) |
| Deceased by 28 days | 3.3 (1031) |
| Missing | (0.1) 44 |
| Missing birth weight | 12.4 (3863) |
| Timing of birth weight measurement (among non-missing birth weight, n=27 253) | |
| 0–5 hours | 17.5 (4775) |
| 6–23 hours | 57.9 (15 773) |
| 24–72 hours | 11.1 (3018) |
| >72 hours | 13.5 (3685) |

Continued

**Table 3** Continued

| | All live births (n=31 116) % (n) |
|---|---|
| Timing missing | 2 |

LMP, last menstrual period; NOMS, Nepal Oil Massage Study.

average a 23 g reduction for measurements taken on days 4–9 and an 857 g average reduction for measurements taken on day 10 or more after delivery (figure 2). On average, babies that did not survive the neonatal period had an imputed weight 324 g less than babies that did, and multiples had an imputed weight 659 g less than singleton babies (online supplemental table 1). There was a small amount of heaping at 4500 g for the measured weights; n=70 babies at exactly 4500 g or a heaping index of 35% (±250 g) (figure 2).

Using the imputed birth weights, we found differences in the SGA prevalence compared with weights taken within 72 hours after delivery (table 4). SGA prevalence using measured weights was 49% (95% CI: 48% to 49%) and 44% (95% CI: 43% to 44%) using imputed weights. The prevalence of LGA babies was similar in absolute terms (4% using the raw birth weights compared with 5% using the imputation). However, in relative terms, this is a 24% increase in the estimate of LGA from the measured to imputed. The proportion of low birthweight babies decreased from 30% (95% CI: 30% to 31%) to 27% (95% CI: 26% to 27%) and the proportion of very low birth-weight babies (<1% of the sample) remained the same.

The sensitivity analysis showed no changes in baby gender or term or preterm delivery, except for very preterm. The proportion of babies born at 28–32 weeks increased significantly (95% CI not overlapping) using the imputed dataset. Also, the neonatal mortality rate increased 109% from 16 deaths to 33 per 1000 live births (table 4). Among the babies with non-missing birth

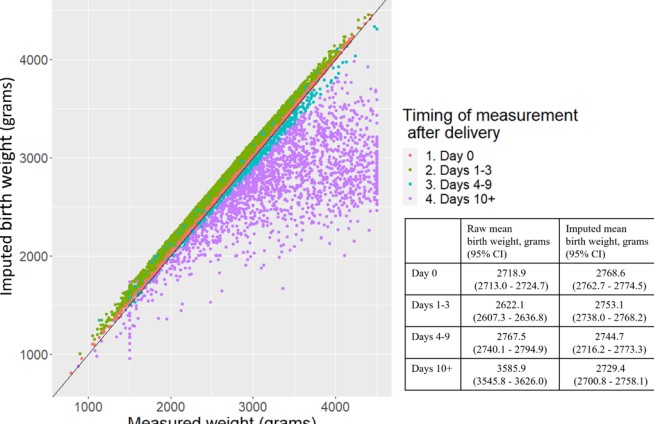

**Figure 2** Comparison of measured weight and imputed weight in grams by the timing of the weight measurement in days after delivery, Nepal Oil Massage Study dataset.

**Table 4** Comparison of low birthweight, small-for-gestational age and large-for-gestational age estimates

| | Measured weight <72 hours | Imputed birth weight | Relative change (%) | Absolute difference |
|---|---|---|---|---|
| **Birth weight (g); % (95% CI)** | | | | |
| Low birth weight (<2500) | 30.1 (29.5 to 30.7) | 26.7 (26.0 to 27.4) | 11.3 | −3.4 |
| Very low birth weight (<1500) | 0.5 (0.4 to 0.6) | 0.5 (0.4 to 0.6) | 0 | 0 |
| **Size-for-gestational age in percentile; % (95% CI)** | | | | |
| Appropriate-for-gestational age (90th–10th centile) | 47.7 (47.1 to 48.3) | 51.6 (50.9 to 52.2) | 8.2 | +3.9 |
| Small-for-gestational age (<10th centile) | 48.5 (47.9 to 49.1) | 43.7 (43.1 to 44.3) | 10.0 | −4.8 |
| Large-for-gestational age (>90th centile) | 3.8 (3.6 to 4.1) | 4.7 (4.5 to 5.0) | 23.7 | +1.0 |
| **Gestational age at delivery (weeks); % (95% CI)** | | | | |
| Term births (37–42) | 74.3 73.7 to 74.9 | 73.3 72.8 to 73.8 | 1.3 | −1.0 |
| Moderate-to-late preterm (32–37) | 13.4 13 to 13.8 | 13.7 13.3 to 14.1 | 2.2 | +0.3 |
| Very preterm (28–32) | 1.3 1.2 to 1.4 | 1.6 1.5 to 1.8 | 23.7 | +0.3 |
| Extremely preterm (24–27) | 0.3 0.2 to 0.3 | 0.4 0.3 to 0.4 | 33.3 | +0.1 |
| Post-term (42–45) | 10.8 10.4 to 11.2 | 11.0 10.6 to 11.3 | 1.9 | +0.2 |
| **Female gender; % (95% CI)** | 48.6 47.9 to 49.2 | 48.1 47.6 to 48.7 | 1.0 | −0.5 |
| **Male gender; % (95% CI)** | 51.5 50.8 to 52.1 | 51.9 51.3 to 52.5 | 0.8 | +0.4 |
| **Neonatal mortality rate; deaths per 1000 live births (95% CI)** | 15.9 14.4 to 17.6 | 33.2 31.3 to 35.3 | 108.8 | 17.3 |
| **Birth weight (in g) among babies with non-missing weight; mean (95% CI)** | | | | |
| Neonatal period survival | 2713.2 2707.8 to 2718.7 | 2775.5 2770.0 to 2781.0 | 2.3 | 62.3 |
| Neonatal death | 2236.9 2171.8 to 2302.0 | 2290.0 2226.1 to 2353.9 | 2.4 | 53.1 |

weights, the mean birth weight measured within 72 hours and the imputed birth weight was not statistically significant among neonatal deaths.

## DISCUSSION

Birth weights measured during the nadir bias estimates of SGA, LGA and low birth weight status. Our recalibration model pulled the weights back to t=0 at delivery, increasing the weights measured at the nadir, and decreasing weights measured after the nadir as the baby begins to surpass birth weight. The imputation also generated birth weight estimates for babies missing a measured weight, many of whom were early neonatal deaths and/ or very preterm babies who died before the first postnatal study visit. In this dataset, these factors resulted in

a reduced prevalence of small, vulnerable newborns. The proportion of babies with SGA is reduced by a relative 10% and low birth weight by a relative 11% when using the recalibrated and imputed birth weights.

Fonseca et al modelled neonatal weight loss among full term, breastfed, singleton Portuguese infants with weight measured within 96 hours.[14] They fit several longitudinal models of weight change, but the goal was to examine the patterns and estimate a nadir, not to extrapolate the infant weights back to a 'true' birth weight. This model was done in a high-income setting with low SGA prevalence and different patterns of facility delivery/caesarean section compared with many LMICs in South Asia.

In the first week after delivery, there is evidence that infants follow a broadly consistent pattern of weight

change. Weight loss 2–3 days postnatally is primarily due to loss of water weight as the cardiopulmonary system readapts for life outside the womb.[15] Even babies born preterm lose weight primarily due to reduced extracellular water volume.[16 17] However, IUGR babies may have a different pattern of weight change. One study of SGA/term and SGA/preterm babies found weight loss was due primarily to catabolism rather than loss of extracellular volume.[18] Another study comparing SGA with AGA, preterm babies found less postnatal diuresis in the SGA babies.[19] In settings with a high prevalence of SGA, population-level estimates of postnatal growth in the first week are likely to be different compared with low SGA prevalence areas—due to different growth patterns in babies with IUGR. This model should be tested in other settings but has the potential to be used for correction of birth weight measured in the days postdelivery, particularly in South Asian LMICs where SGA prevalence is high compared with other regions.

The main strength of this study is that we had a very large dataset with a relatively small proportion of missing covariates. In both the CHX and NOMS studies, we have high-quality gestational age at birth and weight was measured on a population-representative sample. The two studies were conducted in the same geographic area, from the same study site, and were managed by the same research organisation, the Nepal Nutritional Intervention Study Sarlahi. Another strength of this study is we were able to use daily, longitudinal weights to build the model of early neonatal growth.

One limitation is the differences between the two datasets. Difference in timing of data collection: CHX (2002–2005) vs NOMS (2010–2017). Babies born during the NOMS data collection may have a different weight change pattern compared with babies born in CHX trial due to secular trends in maternal health and nutritional status. The per cent of women delivering at home has declined from CHX to NOMS, primarily due to the cash incentive programme in Nepal that pays women to deliver in a facility.[20] However, the median birth weights (table 1) and the neonatal mortality rates are similar (30 neonatal deaths per 1000 live births in CHX and 31 per 1000 live births (personal communication, Joanne Katz, Johns Hopkins Bloomberg School of Public Health, November 2021) in NOMS).[9]

All infants in the CHX subsample of longitudinal weights survived through the neonatal period and were singleton births. This is an important difference in the underlying population of the two samples (CHX vs NOMS) since multiples typically have lower birth weight and very SGA/preterm/low birthweight babies are more likely to die in the few days after delivery. The imputed and measured mean birth weights by survival status were similar, among the babies with non-missing birth weight, indicating that the imputation model performed well (table 4).

Another limitation is that the birth weights are not missing at random in NOMS. More than half the infants died in the NOMS study before their weight could be measured, and more preterm babies did not have a weight measurement (likely also due to early neonatal death). We found a higher proportion of very preterm babies in the sample with the imputed birth weights and a much higher neonatal mortality rate (table 4). We found infants of women with more education and wealth were missing birth weight, perhaps because they were more mobile and more likely to move out of the study area, or more likely to deliver at a facility and returned home after some days. Nulliparous women were also more likely to have no weights measured on their infants. This may be due to the custom of women delivering their first baby at their parents' home. We included these covariates in the model to adjust for these differences but there is likely still residual confounding.

## CONCLUSION

Using weights measured at or around the nadir—as typically occurs during community-based studies due to the logistics of quickly transporting data collection teams—overestimates the prevalence of SGA and low birth weight. We found a lower prevalence when using birth weight imputed to the time of delivery compared with weight taken within 72 hours postdelivery in a sample of deliveries in rural Nepal—an area where less than half of women deliver at facilities, high levels of exclusive breast feeding and only half of babies are born with an appropriate weight for gestational age. Other studies using birthweight measurements taken from infants born at home should consider a similar imputation model to generate more accurate SGA/low birthweight status at the population level.

Overall, more research on improving estimates of SGA is needed. This model may be generalisable to other South Asian LMIC countries but may not be in LMICs with lower SGA prevalence, such as sub-Saharan Africa. Cohort studies with birthweight measurements should consider measuring longitudinal, early neonatal weight change since data in more diverse settings are needed, especially if a similar birthweight recalibration and imputation is planned. A recalibration model may also be clinically useful to better discriminate smaller babies, improving specificity of care. Finally, we note this model is restricted to live births since neither study measured weights on stillbirths. It would be culturally inappropriate to ask families to retain the fetal remains until study staff could arrive for weight measurement. However, IUGR may be an important cause of fetal death in this area and excluding stillbirths will likely underestimate the prevalence of SGA. Our study could not address this but including stillbirths in global estimates is critical to understanding the burden of IUGR and further work is needed to determine whether stillbirth weights may be imputed.

**Author affiliations**
[1]International Health, Johns Hopkins University Bloomberg School of Public Health, Baltimore, Maryland, USA
[2]Global Health, George Washington University Milken Institute School of Public Health, Washington, District of Columbia, USA
[3]Nepal Nutrition Intervention Project Sarlahi, Kathmandu, Nepal

**Correction notice** The article has been corrected since it was published online. Minor typos in one of the formulas have been corrected in Box 1.

**Acknowledgements** We would like to acknowledge Professor Anne CC Lee at Harvard Medical School and Professor Robert Black at Johns Hopkins Bloomberg School of Public Health for their technical contributions, and the study staff and participants at the Nepal Nutritional Intervention Project Sarlahi site.

**Contributors** JK conceptualised and designed the study, designed the data collection protocols, developed the analytic plan and statistical models and reviewed and revised the manuscript. LCM conceptualised and designed the study, designed the data collection protocols, revised data collection tools and supervised data collection, developed the analytic plan and statistical models and reviewed and revised the manuscript. JMT conceptualised and designed the study, designed the data collection protocols and reviewed and revised the manuscript. SKK revised data collection tools and supervised data collection and reviewed and revised the manuscript. SLZ developed the analytic plan and statistical models and reviewed and revised the manuscript. DM and SS contributed significantly to the analysis and critically reviewed and revised the manuscript for important intellectual content. EAH developed the analytic plan and statistical models, carried out the analysis and drafted the initial manuscript. All authors reviewed and approved the final manuscript as submitted and agreed to be accountable for all aspects of the work in ensuring that questions related to the accuracy or integrity of any part of the work are appropriately investigated and resolved. EAH is the guarantor and accepts full responsibility for the work and/or the conduct of the study, had access to the data, and controlled the decision to publish.

**Funding** This work was supported by the National Institute for Child Health and Human Development (R01HD092411 and R01HD060712), the Bill & Melinda Gates Foundation (OPP1084399), National Institutes of Health (810-2054) and Cooperative Agreements HRN-A-00-97-00015-00 and GHS-A-00-03-000019-00 between Johns Hopkins University and the Office of Health and Nutrition, US Agency for International Development.

**Competing interests** None declared.

**Patient and public involvement** Patients and/or the public were not involved in the design, or conduct, or reporting, or dissemination plans of this research.

**Patient consent for publication** Not applicable.

**Ethics approval** This is a secondary data analysis study of two trials for which institutional review board approvals were obtained from the Johns Hopkins Bloomberg School of Public Health (FWA00000287) and Nepal Health Research Council (FWA 00000957). All human subjects in both studies gave their informed consent to participate. This secondary data analysis was considered exempt by the Johns Hopkins Bloomberg School of Public Health institutional review board and conforms to the principles embodied in the Declaration of Helsinki.

**Provenance and peer review** Not commissioned; externally peer reviewed.

**Data availability statement** Data are available on reasonable request.

**ORCID iDs**
Elizabeth A Hazel http://orcid.org/0000-0002-9176-3278
Luke C Mullany http://orcid.org/0000-0003-4668-9803
Scott L Zeger http://orcid.org/0000-0001-8907-1603
Seema Subedi http://orcid.org/0000-0002-6360-3998
James M Tielsch http://orcid.org/0000-0002-1151-060X
Joanne Katz http://orcid.org/0000-0002-5997-7823

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
