## [Reviewer comments · BMJ Open]

ARTICLE DETAILS

TITLE (PROVISIONAL)	Development of an imputation model to recalibrate birthweights measured in the early neonatal period to time at delivery and assessment of its impact on size-for-gestational age and low birthweight prevalence estimates: a secondary analysis of a pregnancy cohort in rural Nepal
AUTHORS	Hazel , Elizabeth; Mullany, Luke C.; Zeger, Scott; Mohan, Diwakar; Subedi, Seema; Tielsch, James; Khatry, Subarna; Katz, Joanne

VERSION 1 – REVIEW

REVIEWER	Rasmussen, Svein Universitetet i Bergen
REVIEW RETURNED	22-Jan-2022

GENERAL COMMENTS	The Introduction section is adequate, but contains reiterations (e.g. «Weights of babies taken during the nadir may bias SGA estimates, overestimating the prevalence»). The explanation of effects on population prevalence of SGA of weighting the baby at the nadir can should be condensed. At the end of the Introduction the aim of the study is clearly formulated («.. to generate estimates of SGA, LGA, and low birth weight using an imputation model that extrapolates infant weights measured post-delivery (around the nadir) to an imputed weight at the time of birth”). The following sentence («In developing ...in South Asia») is a method issue and can be deleted. The last sentence of the section can be moved up. The Methods section would have been easier to follow if the tree steps described, including the regressions, were described at one place, avoiding reiterations, and not spread throughout the section. In Table 1, does “Median gestational age” refer to gestational age at delivery? At the end of page 7, below “Descriptive analysis”, It is unclear if the model was a Cox regression and the outcome and exposures should be clearly described. The models (equations) used in each consecutive step should be given. Although some of the authors presumably are native English speaking, I think that the language is not optimal, e.g. “Recall of LMP was relatively short as it was ascertained during visits at five-week intervals for pregnancy surveillance of women of childbearing age”. Was Stata used in all statistical calculations? The description of the results is easier to read, and is better written, than the Methods section.
--

	Do the lines around the probability graph in Figure 1 indicate 90% confidence interval. In Table 3, does “>72 hours” below “Timing of birthweight measurement (among non-missing birthweight)” include not measured? The Discussion could have been better structured. A Discussion section usually starts with a brief statement of the principal results, but the first paragraph contains pathophysiologic and future study issues. The Conclusions should be condensed. Do the authors mean “remains” and not “remained” in “It would be culturally inappropriate to ask families to retain the fetal remained until study staff could arrive for weight measurement»? (Last paragraph).
--	--

REVIEWER	Ghosh, Rakesh University of California San Francisco, Institute for Global Health Sciences
REVIEW RETURNED	30-Mar-2022

GENERAL COMMENTS	This is a well-written manuscript. It reports the extent of misclassification that might happen if weight measurement is not done immediately after birth and the delayed first measurement is used as birth weight. This is important contribution towards understanding the situation in LMICs where home deliveries continue to occur quite frequently, and birth weight measurement are often delayed. Additional details will increase clarity of the manuscript and help methods minded readers get a clearer picture. I am providing my comments by line numbers so the major and minor comments are mixed. Line 73: Not just USG at any time during pregnancy but USG in the first trimester, which has higher accuracy in predicting the date of conception. Line 80: Please state 7-8% of what? Birth weight? Same for line 81. Line 83: Six hours is still quite a long time for facility deliveries and in which setting? Is there a good reference that covers a specific region or LMIC countries? More context and appropriate citations to this sentence will be useful. Line 104: How frequently followed up within the 28-day period since birth? Table 1: Suggestion for a left column item name “Median time of first weight measurement after birth”. What proportion of the two cohorts are home deliveries, especially for the CHX study? Any reason why you presented the medians in table 1 not the mean. Are those not normally distributed? Suggest adding at least one measure of variation (IQR with median, SD with the mean) and/or preferably the range if the tail is long. Line 115: Is that a systematic random sample of births, for the longitudinal sub study? Are those selected, a mix of home and facility deliveries? I see you mention weight measured within 6 hours. If 457 is the n for the longitudinal sub study, what is the full sample for the parent study. How does the birth weight distribution of this sub study compare with the distribution of the parent study?
--

Line 122: Please explain your choice for excluding neonatal deaths in the CHX sample. What could be the consequence, in terms of bias, for excluding those? Given that PTB infants and those who are preterm and IUGR are at the highest risk of death, isn't this truncating the tail in a selective manner. Line 278 says that all infants survived in the sub sample, but this sub sample is selected from a larger study. It will be surprising if there were no deaths in the parent study. Needs more information and clarification on this point.

A sentence explaining your rationale (e.g., granular data etc.) for using the CHX cohort as your training sample will be useful addition.

Line 127: Please provide the range of time for that first postnatal study team visit in the NOMS study. This is related to my suggestion above for Table 1.

Line 130: Says same scale but table 1 last row says CHX study used two scales.

Line 131: If a woman delivered in a facility, I would imagine, birth weight was measured at the time of birth, was it not? If not, please clarify. If measured, did you not use the weight as birth weight. I am not understanding the significance of this sentence but the point it was stated suggests that it is of vital/some importance.

Line 136: For a small group of interested readers, it will be helpful if you write out this function/ equation either in the main text or in a supplement. It will also allow someone to use those if needed.

Line 140: please include probability of return to birth weight within 10 days. Otherwise, the sentence keeps me guessing until I read the next sentence.

Line 147: I am unable to understand because you say "... by estimating, then simulating ...". Are you estimating for case 1 and simulating for case 2? If so, please add "respectively." Suggest using the word impute as you have done later on, throughout the manuscript. Please write the equation for Steps 1 and 2, which will help understand the difference between the models obtained from the two datasets.

Line 158: Is a dichotomous PTB better predictor than continuous gestational age? Are no measures of pre-pregnancy weight or pregnancy weight gain available or are they not considered because they are not good predictors in this dataset or because these are events primarily related to the first few weeks? This additional model fitting information will be helpful for those interested. What are the model fit statistics such as the R2 etc. and how did this change when you included or excluded the variables of interest? Table A1 is very helpful. If R2 is not available for mixed effect model, you can present the quasi-log likelihood, AIC etc. comparing the models. Reference for parity is a novel way of presenting, so I guess you are saying for a nulliparous woman it is "0" etc.

Lines 172-173: In other words, you used the model developed from the CHX data, and predicted birth weight for the observed NOMS data, is that correct? Your use of words like "applied" and "improved" in this sentence is good, but more detail-oriented ones would love to understand what exactly they mean. When you write "improved",

	how did you determine it improved? Figure 2 is great and somewhat makes this case (reconsider the marker color green and blue, overlap is difficult to read with these colors). The paragraph starting in line 222, please provide numerical estimates (like the n, median, IQR and the range for the categories in figure 2). Line 226, you provided the average but how big was the range and is it based on all 1031 deaths. A table comparing statistics of births where weight was measured on day 0, 1 etc. would tell the readers how well the model is performing. Is the model performing equally well for all weight categories, for those died and for those when weight was first measured on day 0, 1, 2 or later? Currently figure 2 tells that may or may not be the case. This point merits inclusion early on in the discussion. In addition to the overall estimation of misclassification as in table 4, consider sensitivity analyses in different sub-groups to examine if the model is performing better in some cases than others. Line 283: What type of bias can this induce and what could be the magnitude? Can you really apply this model to all births or only to those who survived in the first week or so? A discussion on this point will be helpful in contextualizing the results. Did you compare only those who died in the NOMS data and is that any different from those who survived. I raise this aspect because your CHX data did not include deaths. If the misclassification is similar among those who died vs. who survived in the NOMS data, it will be assured that your CHX data-based model holds good for both groups. Whilst one important strength is to have two independent datasets from the same population, is it limiting generalizability? Could another population be used for prediction and comparison? Figure 1. A footnote to this figure about censoring will make this table more interpretable and stand alone. Otherwise, just from this figure, I would read that those who did not return to their birth weight never attained that weight and most likely died. I hope this helps the authors in obtaining a different perspective on their manuscript.
--	---

VERSION 1 – AUTHOR RESPONSE

Reviewer: 1

Dr. Svein Rasmussen, Universitetet i Bergen Comments to the Author:

8. The Introduction section is adequate, but contains reiterations (e.g. «Weights of babies taken during the nadir may bias SGA estimates, overestimating the prevalence»). The explanation of effects on population prevalence of SGA of weighting the baby at the nadir can should be condensed.

Good point and removed some statements in paragraph two and three of the introduction to be more concise.

9. At the end of the Introduction the aim of the study is clearly formulated («.. to generate estimates of SGA, LGA, and low birth weight using an imputation model that extrapolates infant weights measured post-delivery (around the nadir) to an imputed weight at the time of birth”).

Thank you, no edits made.

10. The following sentence («In developing ...in South Asia») is a method issue and can be deleted.

Removed sentence

11. The last sentence of the section can be moved up.

Agreed and moved to second sentence of the intro section, ln. 76.

12. The Methods section would have been easier to follow if the tree steps described, including the regressions, were described at one place, avoiding reiterations, and not spread throughout the section.

Made multiple revisions throughout the Method section, specifically the Predicting birthweight section(beginning on ln. 170) aiming to make it more clear and concise.

13. In Table 1, does “Median gestational age” refer to gestational age at delivery?

Yes, added clarification to table 1

14. At the end of page 7, below “Descriptive analysis”, It is unclear if the model was a Cox regression and the outcome and exposures should be clearly described. The models (equations) used in each consecutive step should be given.

We plotted a survival curve to show the probability of return to birthweight (revised to clarify). We didn't include any grouping or adjust the probability using covariates so KM or Cox were not used. Revision made on lns. 162-165. We added equations and notation to the methods sections as well to clarify.

15. Although some of the authors presumably are native English speaking, I think that the language is not optimal, e.g. “Recall of LMP was relatively short as it was ascertained during visits at five-week intervals for pregnancy surveillance of women of childbearing age”.

Revised sentence to make it more clear, lns 140-143. Also carefully reviewed paper for conciseness and clarity and made edits throughout in TC.

16. Was Stata used in all statistical calculations?

We used R to fit the model for the training data and generate the m=5 dataset. Apologies for this oversight. It was included in earlier iterations of the paper but was dropped at some point. See ln. 227

17. The description of the results is easier to read, and is better written, than the Methods section.

Thank you, no edits made.

18. Do the lines around the probability graph in Figure 1 indicate 90% confidence interval.

Yes, 95% confidence interval, we will add that information to the figure 1 caption.

19. In Table 3, does “>72 hours” below “Timing of birthweight measurement (among non-missing birthweight)” include not measured?

No, it the proportions do not include babies with missing birthweight. Revised table 3 and added the

sample size for that category to clarify.

20. The Discussion could have been better structured. A Discussion section usually starts with a brief statement of the principal results, but the first paragraph contains pathophysiologic and future study issues.

Please see edits made throughout the Discussion section in TC.

21. The Conclusions should be condensed.

Please see edits made throughout the Conclusion section in TC.

22. Do the authors mean “remains” and not “remained” in “It would be culturally inappropriate to ask families to retain the fetal remained until study staff could arrive for weight measurement»? (Last paragraph).

Thank you, typo corrected.

Reviewer: 2

Dr. Rakesh Ghosh, University of California San Francisco Comments to the Author:

This is a well-written manuscript. It reports the extent of misclassification that might happen if weight measurement is not done immediately after birth and the delayed first measurement is used as birth weight. This is important contribution towards understanding the situation in LMICs where home deliveries continue to occur quite frequently, and birth weight measurement are often delayed. Additional details will increase clarity of the manuscript and help methods minded readers get a clearer picture. I am providing my comments by line numbers so the major and minor comments are mixed.

23. Line 73: Not just USG at any time during pregnancy but USG in the first trimester, which has higher accuracy in predicting the date of conception.

Revised to clarify this. Ln. 86-87

24. Line 80: Please state 7-8% of what? Birth weight? Same for line 81.

Yes, relative reduction in weight compared to the birthweight. Added this to both sentences. Ln. 95-96

25. Line 83: Six hours is still quite a long time for facility deliveries and in which setting? Is there a good reference that covers a specific region or LMIC countries? More context and appropriate citations to this sentence will be useful.

Good point, and revised this to “typically measured shortly after delivery” since it may vary by setting. Ln. 98-99

26. Line 104: How frequently followed up within the 28-day period since birth?

Neonates were followed up on 1, 3, 7, 10, 14, 21 and 28 of days of age. Added this information to the manuscript as well. Ln. 123-124

27. Table 1: Suggestion for a left column item name “Median time of first weight measurement after

birth". What proportion of the two cohorts are home deliveries, especially for the CHX study? Any reason why you presented the medians in table 1 not the mean. Are those not normally distributed? Suggest adding at least one measure of variation (IQR with median, SD with the mean) and/or preferably the range if the tail is long.

- a) Revised the left column item name as suggested and added uncertainty measurements to table 1.
- b) Added a row for facility delivery to table 1.
- c) The means and medians of all measurements in table 1 are similar suggesting a normal distribution except for the timing of first measurement in NOMS. The birthweights in the CHX subsample were taken daily and stopped at 10 days. The NOMS weight measurements were taken over a longer period of time and some were taken up to weeks after delivery resulting in right skewed data. We added a line to table 1 also showing the mean for timing of first weight measurement. (Table below not included)

CHX NOMS

Mean / Median Mean / Median

Gestational age at delivery (weeks) 39.45 / 39.7 39.19 / 39.4

Birthweight (non-imputed, grams) 2710 / 2706
2770.35 / 2740

Time of first weight measurement after birth (hours) 4.03 / 4.1
89.1 / 15.2

28. Line 115: Is that a systematic random sample of births, for the longitudinal sub study? Are those selected, a mix of home and facility deliveries? I see you mention weight measured within 6 hours. If 457 is the n for the longitudinal sub study, what is the full sample for the parent study. How does the birth weight distribution of this sub study compare with the distribution of the parent study?

- a) Revised to read purposely selected. It was not a random selection. Ln. 137
- b) Table with the median/mean birthweight from the full CHX sample is below (n=22,762) and the distribution appears to be comparable. Suggesting no additional information added about the full CHX sample since that information is available in the CHX study paper that is cited.

CHX subsample CHX total sample (n=22,762)

Median birthweight (non-imputed, grams) 2706

IQR: 2480 – 2986 2700

IRQ: 2416-3000

Mean: 2703.6

SD: 443.4

29. Line 122: Please explain your choice for excluding neonatal deaths in the CHX sample. What could be the consequence, in terms of bias, for excluding those? Given that PTB infants and those who are preterm and IUGR are at the highest risk of death, isn't this truncating the tail in a selective manner. Line 278 says that all infants survived in the sub sample, but this sub sample is selected from a larger study. It will be surprising if there were no deaths in the parent study. Needs more information and clarification on this point.

Good point, add clarification that early neonatal deaths would not be included in this subsample – by definition – since the baby would need to survive the ten days in this sample in order to have their weight measured. (Ln. 143-144) There were neonatal deaths in the parent study and more

information on the parent study is available at the cited paper.

Babies with a first weight measured within 6 hours were included in the substudy (not a random selection). It is possible that some of those babies may have later died and were then dropped because they didn't have a weight for each of the 10 days. Mortality was not originally captured for this substudy (it was captured for the parent study). So agreed it is not fully representative of the population and those that died would have had a lower birthweight but the regression accounts for this. Added a column to Annex Table 1 that shows the outputs for the regression on the NOMS data.

We suggest removing the number of neonatal deaths in table 1 since the aim of that table is to compare the CHX subsample with NOMS. And instead compare the mortality rates in the two studies (including the whole sample for CHX as you suggest). The mortality rates are given in the limitations section (Ln 343-345.) I also add "early" to Ln.144 to clarify that early neonatal deaths would have been excluded.

30. A sentence explaining your rationale (e.g., granular data etc.) for using the CHX cohort as your training sample will be useful addition.

Added suggested sentence: We selected these data to model early neonatal weight change since it drew from the same population but earlier timepoint than the NOMS study. (Ln.145-146)

31. Line 127: Please provide the range of time for that first postnatal study team visit in the NOMS study. This is related to my suggestion above for Table 1.

The baby's birthweight was measured at the first postnatal visit so will be the same data give in table 1 for "Time of first weight measurement after birth (hours)". Revised sentence on Ln 150 to clarify.

32. Line 130: Says same scale but table 1 last row says CHX study used two scales.

Yes, two different types of scales were used in the CHX trial. Originally, we though the same scale was used but then realized later two scales were used for CHX. We missed updating this in the body of the text. Revised on Ln. 154

33. Line 131: If a woman delivered in a facility, I would imagine, birth weight was measured at the time of birth, was it not? If not, please clarify. If measured, did you not use the weight as birth weight. I am not understanding the significance of this sentence but the point it was stated suggests that it is of vital/some importance.

Thanks and it's a minor point. The birthweight would have been measured if a women delivered at a facility, but for the study, the study team would have re-measured the baby and inputted that as the birthweight. There were concerns about the data quality and inconsistency of the birthweight data at the facility. I removed this distinction and just mentioned the study teams measured all birthweights. Ln. 154-155

34. Line 136: For a small group of interested readers, it will be helpful if you write out this function/ equation either in the main text or in a supplement. It will also allow someone to use those if needed.

Thanks, agreed that would be helpful. Wrote our mixed multi-level model equation used to predict birthweight for CHX and NOMS datasets and then the notation of how the parameters from CHX were applied to NOMS to generated the recalibrated birthweight. Ln. 182 for equation, 183-186 for CHX textual description, and 200-201 for a shorter description of the model applied to the NOMS.

35. Line 140: please include probability of return to birth weight within 10 days. Otherwise, the sentence keeps me guessing until I read the next sentence.

Added this Ln. 164

36. Line 147: I am unable to understand because you say “.... by estimating, then simulating” Are you estimating for case 1 and simulating for case 2? If so, please add “respectively.” Suggest using the word impute as you have done later on, throughout the manuscript. Please write the equation for Steps 1 and 2, which will help understand the difference between the models obtained from the two datasets.

Revised to clarify throughout this section in TC, added MLM equation and Box 1 with the exact steps.

37. Line 158: Is a dichotomous PTB better predictor than continuous gestational age? Are no measures of pre-pregnancy weight or pregnancy weight gain available or are they not considered because they are not good predictors in this dataset or because these are events primarily related to the first few weeks? This additional model fitting information will be helpful for those interested. What are the model fit statistics such as the R2 etc. and how did this change when you included or excluded the variables of interest? Table A1 is very helpful. If R2 is not available for mixed effect model, you can present the quasi-log likelihood, AIC etc. comparing the models. Reference for parity is a novel way of presenting, so I guess you are saying for a nulliparous woman it is “0” etc.

1. Revised to clarify it's continuous gestational age at delivery with a spline at 40 weeks Ln. 185

2. Thanks for the suggestion. Added in a Table A2 that shows the AIC of a few different models of maternal-level covariates. The full model had the best fit and the model excluding maternal education was second best (in line with what table A1 shows – education by year having a minimal impact on weight). See text on lns 163-265

38. Lines 172-173: In other words, you used the model developed from the CHX data, and predicted birth weight for the observed NOMS data, is that correct? Your use of words like “applied” and “improved” in this sentence is good, but more detail-oriented ones would love to understand what exactly they mean. When you write “improved”, how did you determine it improved? Figure 2 is great and somewhat makes this case (reconsider the marker color green and blue, overlap is difficult to read with these colors).

a. Changed the terminology throughout the paper and the title to recalibration and imputation to indicate we “recalibrated” the weights to time=0 and then used those to impute missing birthweights. We had been using the term “imputation” for the entire process and we agree this is more clear.

b. Revised figure 2 as suggested.

39. The paragraph starting in line 222, please provide numerical estimates (like the n, median, IQR and the range for the categories in figure 2). Line 226, you provided the average but how big was the range and is it based on all 1031 deaths. A table comparing statistics of births where weight was measured on day 0, 1 etc. would tell the readers how well the model is performing.

Great suggestion. Added this table as part of the overall figure 2 so readers can see the distribution and the means and added the numerical estimates in the text as suggested.

40. Is the model performing equally well for all weight categories, for those died and for those when weight was first measured on day 0, 1, 2 or later? Currently figure 2 tells that may or may not be the case. This point merits inclusion early on in the discussion. In addition to the overall estimation of misclassification as in table 4, consider sensitivity analyses in different sub-groups to examine if the model is performing better in some cases than others.

Great suggestion: we looked at whether using the imputed sample changes the proportion of gender or term/preterm. If it does, that would mean babies are being systematically included (or excluded) with the imputation and they all appear to be the same except for the neonatal mortality rate and the proportion of babies born very preterm increases. This is because the imputation generates a birthweight for babies that died before a birthweight could be measured (half the neonatal deaths in this sample). Thus imputation includes these deaths that were previously excluded. We also added a category for very low birthweight to examine any differences. See additions to Table 4. See Ins 298-303 and 354-356.

41. Line 283: What type of bias can this induce and what could be the magnitude? Can you really apply this model to all births or only to those who survived in the first week or so? A discussion on this point will be helpful in contextualizing the results. Did you compare only those who died in the NOMS data and is that any different from those who survived. I raise this aspect because your CHX data did not include deaths. If the misclassification is similar among those who died vs. who survived in the NOMS data, it will be assured that your CHX data-based model holds good for both groups.

Good idea. We compared the mean measured and imputed birthweight by survival status among those babies with a non-missing birthweight and the means are similar (not statistically significantly different by comparing the 95% CIs). This is evidence the recalibration process is working well. See revised on Table 4 and Ins 301-303 and 349-351

42. Whilst one important strength is to have two independent datasets from the same population, is it limiting generalizability? Could another population be used for prediction and comparison?

Yes, and we aim to test this with a longitudinal early neonatal weight change dataset in Sub-Saharan Africa as well. This is mentioned in Conclusions Ins. 373-375

43. Figure 1. A footnote to this figure about censoring will make this table more interpretable and stand alone. Otherwise, just from this figure, I would read that those who did not return to their birth weight never attained that weight and most likely died.

Thanks, added to figure 1

VERSION 2 – REVIEW

REVIEWER	Rasmussen, Svein Universitetet i Bergen
REVIEW RETURNED	19-May-2022
GENERAL COMMENTS	In the title, an impotent comma should be inserted between «delivery» and «its impact». In the Abstract, «LBW prevalence» is hardly an outcome («The primary outcomes are the relative and absolute differences in SGA and LBW prevalence comparing the imputed and measured birthweights»), but «LBW» is. Below Results in the Abstract «by 11 %» in «Low birthweight is

	reduced by 11 %, from 30% (95% CI: 30-31%) to 7% (95% CI: 26-27%)» is unnecessary. Likewise, in «The proportion of babies born large-for-gestational age increased from 4% (95% CI: 3.6-4.1) to 5% (95% CI: 4.5-5.0) or a relative increase of 24%» «or a relative increase of 24%» is unnecessary. $100\% \times (5-4)/4 = 25\%$, not 24%, but may be I misunderstand. In the Introduction in «The etiology of babies born with IUGR is likely distinct from those born preterm, and perhaps even different from babies born early with IUGR», «The etiology of babies born with IUGR» could simplified to «The etiology of IUGR», because babies cannot have an etiology. In Table 1 mean «Time of first weight measurement after birth (hours)» in the study data was 89.1SD: 275.42 which is quite different from the median. Is this discrepancy due to strongly skewed data of an error?
--	---

REVIEWER	Ghosh, Rakesh University of California San Francisco, Institute for Global Health Sciences
REVIEW RETURNED	06-Jun-2022

GENERAL COMMENTS	The authors have adequately addressed the comments and suggestions. A few monitor points to consider. *Consider a punctuation in the title after the word delivery, if that is allowed by the Journal format. *Check the sentence on design in the abstract. *Define X1 and Xp in the first equation. *I agree with Reviewer 1 about the structure of the first discussion paragraph. It can be separated into two paragraphs. First summarizing and interpreting the results, then contextualizing the results of this study with other relevant ones. *Overall, I imagine the text will be vetted in the proofing process. A few places need attention (highlighted in the attached pdf). I have no further suggestion. I believe the manuscript will be good contribution to literature.
--

VERSION 2 – AUTHOR RESPONSE

Reviewer: 1

Dr. Svein Rasmussen, Universitetet i Bergen Comments to the Author:

In the title, an impotent comma should be inserted between «delivery» and «its impact».

Revised title

In the Abstract, «LBW prevalence» is hardly an outcome («The primary outcomes are the relative and absolute differences in SGA and LBW prevalence comparing the imputed and measured birthweights»), but «LBW» is.

Line 40: Revised to read: The primary outcomes are size-for-gestational age and LBW.

Below Results in the Abstract «by 11 %» in «Low birthweight is reduced by 11 %, from 30% (95% CI: 30-31%) to 7% (95% CI: 26-27%)» is unnecessary.

Line 44: Removed by 11% and other references to the relative change in the abstract.

Likewise, in «The proportion of babies born large-for-gestational age increased from 4% (95% CI: 3.6-4.1) to 5% (95% CI: 4.5-5.0) or a relative increase of 24%» «or a relative increase of 24%» is unnecessary. $100\% \times (5-4)/4 = 25\%$, not 24%, but may be I misunderstand.

Line 45: Removed relative increase of 24%. The difference between the 24% and 25% is rounding.

Table 4 is more precise: $(4.7-3.8)/3.8 = 23.7\%$

In this manuscript, we used one decimal point precision for the tables but rounded when referencing in the text to improve readability. Please let me know if the journal prefers another method.

In the Introduction in «The etiology of babies born with IUGR is likely distinct from those born preterm, and perhaps even different from babies born early with IUGR», «The etiology of babies born with IUGR» could be simplified to «The etiology of IUGR», because babies cannot have an etiology.

Line 79: Revised as suggested

In Table 1 mean «Time of first weight measurement after birth (hours)» in the study data was 89.1SD: 275.42 which is quite different from the median. Is this discrepancy due to strongly skewed data of an error?

Yes, the data are right-skewed due to a small number of babies with postnatal visits a couple of weeks after delivery. Added the sentence below to clarify this for other readers.

Line 133: The distribution of visit timing is skewed right since 8% of the babies had their first postnatal visit after day 10 (table 1).

Reviewer: 2 [NOTE FROM THE EDITORS: PLEASE ALSO SEE ATTACHED PDF FOR HIGHLIGHTS FROM REVIEWER - NOTE THAT ONE HIGHLIGHT IS IN THE CLEAN MANUSCRIPT BUT MOST ARE IN THE TRACKED CHANGES VERSION] Dr. Rakesh Ghosh, University of California San Francisco Comments to the Author:

The authors have adequately addressed the comments and suggestions. A few minor points to consider.

*Consider a punctuation in the title after the word delivery, if that is allowed by the Journal format.

Revised title

*Check the sentence on design in the abstract.

Line 28: Revised: We developed and applied a model that recalibrates weights measured in the early neonatal period to time=0 at delivery and uses those recalibrated birthweights to the multiply impute for babies with missing birthweight.

*Define X_1 and X_p in the first equation.

Line 159: In the formula, changed $X \rightarrow x$ to differentiate the fixed effect X_s (X) and random effects X_s (x).

Added detail to these sentences:

Line 161: The fixed effects ($X_{1ij} \dots X_{pij}$) included parity, maternal education/age, infant sex, and gestational age at birth in weeks with a spline at 40 weeks.

Line 163: The random effects within baby include intercept (u_{0j}) and random slope for age ($u_{ij} x_{(1ij)}$).

*I agree with Reviewer 1 about the structure of the first discussion paragraph. It can be separated into two paragraphs. First summarizing and interpreting the results, then contextualizing the results of this study with other relevant ones.

I didn't see this comment from Reviewer 1 but it's a good suggestion. Split the first paragraph of the Discussion and added a few additional sentences summarizing the main findings. (Starting on line 268).

*Overall, I imagine the text will be vetted in the proofing process. A few places need attention (highlighted in the attached pdf).

Thanks for the suggestions! Revised in TC to fix typos or improved clarity in the highlighted sections.

I have no further suggestion. I believe the manuscript will be good contribution to literature.